# The Numerical and Experimental Investigation of Particle Size Distribution Produced by an Electrical Discharge Process

**DOI:** 10.3390/ma14020287

**Published:** 2021-01-08

**Authors:** Faming Lin, Yifan Liu, Xianglong Li, Congqiang Bai

**Affiliations:** 1School of Mechanical Engineering, Southwest Jiaotong University, Chengdu 610065, China; linfaming@swjtu.edu.cn; 2The Yangtze Delta Region Institute (Huzhou), University of Electronic Science and Technology of China, Chengdu 610065, China; 3School of Mechanical Engineering, Sichuan University, Chengdu 610065, China; 4Beijing Sheenline Co., Ltd., Beijing 100089, China; baicongqiang001@sdic.com.cn

**Keywords:** electrical discharge, thermal fragmentation, size distribution

## Abstract

The vague influence of thermal action of electrical discharge on size distribution of metallic powders hinders the adjustability of powder quality. Due to the small gap, short discharge on-time, uncertain discharge point, and strong light interference, direct observation of preparation is difficult to carry out. Herein, the multi-physics coupling finite element method (FEM) was applied to numerically investigate the relationship between size distribution and thermal action. Through modeling of thermal transformation and heat distribution on the surface of electrode, temperature of the electrode is found to be decided by the latent heat and the distribution of heat point obeys the normal distribution, which corresponds with experimental results. Finally, the vapor film to droplet fragmentation theory was proposed to explain the phenomenon of normal distribution. The research results provide theoretical support for the preparation of metallic powder by electrical discharge, and also play a guiding role in optimizing the process parameters in the actual preparation process to tune the size distribution.

## 1. Introduction

Metallic powders have been rapidly increasing in various fields such as science and engineering areas because of their remarkable physical and chemical properties compared to their corresponding bulk materials. Particularly, metallic micropowders have been broadly used in some applications such as additive manufacturing (3-D printing) [1], powder metallurgy [2], and injection molding [3]. The spark discharge method is a physical method for preparing micro-nano metal powder developed in recent years. Compared with the traditional chemical method of metal powder preparing, it has the advantages of green environmental protection, a simple process, easy implementation, and low production cost [4]. Generally speaking, the powder prepared by the spark discharge method tends to have a wider particle size distribution; especially, the metal powder prepared in a liquid environment is easier to find a wider particle size distribution [5,6,7]. Ag nanoparticles were synthesized by arc discharge in a solution such as H_2_O, ethanol (C_2_H_5_OH), and water mixed aqueous solution of polyvinyl pyrrolidone (PVP) with a corresponding average particle size of 8, 10, and 13 nm [8,9]. Spherical Zr nanoparticles are synthesized via electrical discharge with average size decreasing from 40 to 22 nm by increasing the current from 40 to 160 A [10]. Nickel (Ni) nanoparticles were synthesized by a pulse electrical discharge between two nickel electrodes in H_2_O with 1% PVA as the stabilizing agent [11]. The alloy powders can be synthesized through two twisted immiscible metals acting as electrodes. S. M. Kim et al. prepared binary metal (Pt–M (M = Cu, Ag, Pd)) nanoparticles by a spark discharge method in H_2_O [12]. D. Delaportas et al. used different electrodes to prepare different metal oxide nanoparticles in H_2_O [13]. η–Al_2_O_3_ nanoparticles were produced in different carrier media such as H_2_O, ethanol (C_2_H_5_OH), and methanol (CH_3_OH). J. A. Jaworski et al. [14] used a low-powered AC electrical arc (current of 5 to 10 A with a low voltage of 10V at about 2 Hz frequency) to generate different kinds of metallic microparticles in H_2_O and paraffin. M. Mardanian et al. [15] used a high voltage AC power supply (current of 60 A with a high voltage of 8.5 kV at about 100 Hz frequency) to create a pulsed spark discharge with a pulse duration of 30 μs to produce CuInSe_2_ nanoparticles in ethanol (C_2_H_5_OH). Tungsten carbide (WC) nanoparticles were prepared by a pulsed discharge of bulk tungsten and graphite rods immersed in liquid-like ethanol (C_2_H_5_OH), kerosene, and H_2_O [16]. In our previous work, Ni micro-nanoparticles were prepared by ultrasound-assisted electrical discharge erosion process in H_2_O and kerosene [17,18,19,20]. The influence of electrical parameters and ultrasound parameters on forming mechanism, size distribution, morphology, phase and hollow ratio of particles was investigated. The nickel powder particle size distribution is related to energy input and the ultrasonic power. However, the results were based on the experimental analysis. The theorical research was not developed to build the relationship between energy input and size distribution.

To reveal the effect of thermal fragmentation on particle size distribution, the finite element method (FEM) was used to model the electrical discharge and the thermal transmission processes. The predictable distribution law is estimated through numerical analysis, which is beneficial to tune the electrical parameters and optimize the size distribution. Controlled experiments were carried out to verify the numerical results and the experimental distribution agreed with the theorical distribution.

## 2. Numerical Methodology and Experimental Method

### 2.1. Model of Gaussian-Distribution Heat Source

The electrical discharge erosion generator consists of two opposing electrodes separated by a gap. The two electrodes can be made of either the same or different materials. 

The whole preparation process starts from applying a voltage between the electrodes to form a plasma channel in the working fluid. Electrons are first emitted from the cathode surface of the electrode, hitting neutral atoms or molecules in the working fluid between the electrodes, and a large number of conductive particles are increased between the electrodes [21]. The electrons then reach the anode to form a plasma channel. In the plasma channel, due to the high-speed movement of electrons, charged ions collide with each other, generating a large amount of heat, melting the electrode, part of the electrode and the working fluid vaporize, and the volume expands rapidly [22]. The resulting pressure gradient causes molten metal and vaporized metal to peel from the electrode surface and enter the working fluid [23].

The experimental device was described in Figure 1a. When the voltage increased high enough to build an electric field between the electrode gap, the free electrons on the cathode are subjected to electrostatic forces. The high-speed electrons then impinge on the anode and cations on the cathode, causing a Joule heat [24]. Actually, the majority of Joule heat is converted into heat energy to sublimate the surfaces of two electrodes [25]. Material removal occurs due to the instant sublimation of the material as well as evaporation due to melting [26,27]. The heat flux generated by the plasma channel is a function of discharge time, efficiency, energy input, and radial position, obeying a Gaussian distribution equation [28]:(1)$$q\left(r\right)=\frac{4.57\eta \mathrm{VI}}{{\pi R}_{\mathrm{pc}}^{2}}\;\exp\left[-4.5{\left(\frac{r}{R_{\mathrm{pc}}}\right)}^{2}\right]$$
where q(r) is the heat flux density at the radius r and the time t; r is the radial distance from the center of the plasma channel; t is time; η is the proportion of energy obtained by the electrode; V is the discharge voltage; I is discharge current; R_pc_ is the plasma channel radius.

To build the numerical thermal model, an electrode model with radius of 250 μm and height of 400 μm was created in Figure 1c. According to the observer of the electrical discharge channel, 200 μm of discharge radius was set [29]. As shown in Figure 1b, a Gaussian heat source model (Equation (1)) was applied on the surface of electrode. Due to the different energy delivery on two electrodes (anode and cathode), it is concluded that the energy delivery to anode is more than cathode. Here, 30% of the energy is delivered to the cathode [30]. The power density distribution generated by the Gaussian heat source during electrical discharge erosion is demonstrated in Figure 1d.

### 2.2. Governing Equation

To analyze the transient and non-linear thermal conduction process, the Fourier heat conduction equation is taken as the governing equation [31].
(2)$$d_{z}{\rho C}_{p}\left[\frac{\partial T}{\partial t}+\vec{\nabla }\cdot \left(\vec{u}\;T\right)\right]=\vec{\nabla }\cdot \left(k\vec{\nabla }T\right)\;+q_{0}$$

In the formula, T is temperature; t is time; ρ is material density; $C_{p}$ is material specific heat capacity; k is material thermal conductivity; $\vec{u}$ is velocity vector; d_z_ is material thickness; $q_{0}$ is heat flux. The Gaussian heat source generated by the discharge is coupled into the energy equation through $q_{0}$.

### 2.3. Latent Heat Analysis

During the entire discharge process, the electrode has undergone phase changes of melting and vaporizing. The corresponding relationship between energy and temperature during each phase changes, the latent heat will be absorbed in each phase change process, resulting in a discontinuous spatial distribution of the temperature of the electrode after the phase change. Therefore, in order to accurately analyze the temperature and phase change process of the electrode during the discharge process, the influence of latent heat should be fully considered in the calculation process. The minimum energy required for melting and gasification per unit volume $∆H_{m}$ and $∆H_{v}$ are [32]:(3)$$∆H_{m}{=C}_{m}\left(T_{m}{-T}_{0}\right){+L}_{m}$$
(4)$$∆H_{v}=C_{m}\left(T_{m}-T_{0}\right)+C_{v}\left(T_{v}-T_{m}\right)+L_{m}+L_{v}$$

In the formula, C_m_ is specific heat capacity in solid state; T_m_ is material melting temperature; T_0_ is material initial temperature; C_v_ is liquid state specific heat capacity; T_v_ is boiling point temperature; L_m_ is latent heat of melting; L_v_ is latent heat of vaporization.

### 2.4. Boundary Conditions

In electrical discharge erosion process, the heat flux generated by the plasma channel simultaneously acts on the electrode and workpiece, thus in each discharge, the particles are formed not only on the surface of workpiece, but also on the surface of electrode. The workpiece material used in this study is Ni, and the thermo-physical properties of Ni are listed in Table 1. 

For other surfaces of electrode, the thermal insulation boundary was defined as
(5)$$\frac{\partial T}{\partial n}=0$$
where n is the normal direction to the boundary.

### 2.5. Material and Experimental Method

The electrical energy was controlled by the numerical control EDM machine E46PM (from Jiangsu Excellent Numerical Control Equipment Co., Ltd., Jiangsu, China) with variable electrical current ranging from 1.5 to 60 A, variable pulse width ranging from 2 to 1200 μs, and variable gap voltage ranging from 30 to 120 V. Servo system was used to control the two electrodes to maintain an optimal distance of micrometer grade. In order to study the effect of power on the particle size distribution of micro-nano metal powders, we used high-purity Ni rods as the electrode and conducted experiments under the same discharge parameters using H_2_O as the processing medium. After the experiment, the prepared working solution was stirred evenly using a glass rod, then centrifuged at a centrifugal speed of 8000 rpm for 15 min using a high-speed refrigerated centrifuge, and finally the centrifuge was dried into loose nickel particles using a vacuum freeze dryer. A laser particle size analyzer was used to measure the particle size distribution of the test sample.

## 3. Results and Discussion

### 3.1. Temperature Distribution

The relationship between the temperature change along the electrode surface (X axis direction in the Figure 1b) and the depth direction (Z direction in the Figure 1b) with time under a constant voltage and current during the discharge process was obtained by numerical calculation. As shown in Figure 2a–d, the center temperature of electrode surface changes from the initial 273 K to 4000 K as time goes from beginning to 12 μs during the discharging process. After 12 μs, the temperature increases sharply to 6500 K at 15 μs, as shown in Figure 2e–f. The temperature on electrode surface in the planar and depth direction were plotted in Figure 2g–h, respectively. Since the expansion rate of discharge plasma channel is much greater than temperature change rate to the electrode material, the temperature change in the electrode surface direction is greater than the depth direction. Due to the concentration of power on the surface of the electrode during the heating process, temperature shows the same characteristics when conducted along the depth of electrode. The temperature curve along the electrode surface and the depth direction shows a wave-like character which is distributed concentratedly in sub-regions with maximum at the center.

### 3.2. Particle Size Distribution

When the discharge voltage is fixed at 45 V and the current is set at 60 A, with a discharge time of 15 μs, the calculation result of the temperature distribution on the surface of the electrode is shown in Figure 3a. During the discharge process, the electrode temperature distribution is not an ideal linear decrease from the center to the outside. As the electrode latent heat absorption time is inconsistent during the melting or vaporization process, the temperature distribution is concentrated in different regions. From Figure 3b, it can be seen that the electrode surface can be divided into three zones: the gasification phase zone in the center, the melting phase zone, and the solid phase zone. Since the energy density of the center is the largest, the time to start vaporization is the earliest, and at the same time the vaporization and melting area in the center is also the largest. 

To verify the relationship between the distribution of particle size and temperature, the diameter of the concentrated zone of the electrode surface temperature was measured (Figure 3b). It can be seen that the gasification area in the center of electrode surface has the largest area but a small number, with a diameter of ~ 40 μm. The area of the intermediate gasification area is medium in number, with a diameter of ~ 20 μm. After calculating the diameter of different gasification areas, the relationship between the diameter of the gasification area and the number distribution is obtained, which is represented by a red dotted line in Figure 3d, which shows that the calculated size distribution is considered to obey normal distribution. The controlled experiment was carried out by the same discharge parameters with voltage of 45 V, current of 60 A and a discharge time of 15 μs. Figure 3c shows the scanning electron microscope (SEM) image of as-prepared metallic powder. Then the experimental results obtained by the laser particle size analyzer is plotted in solid line in Figure 3d. It is concluded that the experimental size distribution of particles corresponds with the numerical size distribution with the same variation tendency.

To compare the distribution of the vaporization area under different discharge parameters, two groups of contrast experiments were modeled and calculated with the constant discharge voltage and time (45 V and 15 μs) when the discharge current of 30 A and 45 A were applied respectively. The electrode surface temperature distributions for the two contrast experiments were plotted in Figure 3e–f, which shows that with the increase of discharge power, the size and number of vaporization areas increase. Hence, optimized discharge parameters play a crucial role in tuning the size distribution.

Although the simulation analysis was utilized to discuss the relationship between thermal action and size distribution of particles, and the current and voltage pair (i.e., 45 V/60 A vs 60 V/45 A) caused the same Gaussian heat source, indicating the same energy density input, it cannot be used to directly reflect the influence of varying electrical parameters on size distribution. Herein, three groups of single factor experiments were designed to further investigate the rule of effect. The first, second and third group studied the influence of voltage, current, and discharge time on particle size distribution separately. The experimental conditions were shown in Table 2.

It can be seen from Figure 4a that with voltage of 45 V, the particle size distribution is more concentrated and mainly distributed around 10 μm. As the voltage increases to 90 V, the particle size distribution is mainly concentrated around 20 μm. The increase of the discharge voltage makes the energy distributed in the electrode increase and the particle size distribution wider. Figure 4b shows that the main particle size does not increase from 15 to 30 A, but the distribution range is wider. This is mainly due to the lower current and the smaller energy of a single discharge, and the accumulation of heat after repeated discharges to form a melting and vaporization area. As the current increases, the melting and vaporization area increases significantly, the particle size range becomes wider, and the overall particle size becomes larger. Figure 4c shows the discharge time has a significant effect on the particle size distribution. At 15 μs, the particles with size around 10 μm account for about 35%. At doubled discharge time, particle size of the maximum proportion increases to 15 μm. With further increase of time, the proportion of the maximum particle size will not increase significantly, but the overall distribution range of the powder will increase. The three groups of experiments all show that the particle size distribution presents a normal distribution with its number distribution, which means that the large particle size and the small particle size account for a small proportion of the total number, and the medium size has the largest ratio. As the discharge power increases, the particle size distribution range becomes wider, and the average particle size becomes larger. Compared with changing the voltage and discharge time, changing the voltage has the most significant effect on the particle size range and average particle size.

### 3.3. Formation Mechanism of Particle During Electrical Discharge 

During the discharge process, with the increase of electrode temperature, the working fluid around the molten droplet presents a boiling phenomenon, and a vapor film appears near the droplet, which separates the droplet from the working fluid and hinders the transmission from moving forward in hot state. At the end of discharge, with decreasing temperature, the gas film becomes thinner and is partially punctured by the pressure pulse generated by the surrounding environment. 

As the temperature of the droplet drops after the end of the discharge, the gas film becomes thinner, and it will be affected by pressure pulses caused by the surrounding environment, and the gas film is partially pierced [34,35]. The direct contact between the working fluid and the droplet causes a strong heat exchange. The time scale of heat transfer is far less than the time of pressure release [36]. A large amount of water vapor is generated. The steam expands explosively and generates a pressure wave in an extremely short time. The droplet will be broken into a large number of small-volume fragments. The smaller volume of droplets solidifies in the working fluid to form small-sized particles. According to the above analysis, in the whole process of crushing the metal droplets, the main driving force for the metal droplets to be broken is the pressure wave generated by the working fluid contacting the droplets due to the collapse of the steam film and rapid vaporization, as shown in Figure 5.

## 4. Conclusions

As one of the cost, green, efficiency, and productivity approaches, the electrical discharge method is widely applied in preparing the metallic powders due to the simple physical phase change from solid to molten and molten to solid. To reveal the effect of thermal fragmentation on particle size distribution, a series of numerical simulations, experimental verifications, and mechanism discussions were carried out.

In the case of numerical calculation, the Gaussian normal thermal distribution was built and calculated by the finite element method (FEM). Through numerical calculation, the thermal action of electrodes was affected by latent heat during the heating process. In detail, the molten material in the molten pool is concentrated according to temperature. The gasification area in the center has the largest area but a small number, with a diameter of ~ 40 μm. The area of the intermediate gasification area is medium in number, with a diameter of ~ 20 μm. The relationship between the diameter of the gasification area and the number distribution is obeyed by the normal distribution, which is corresponding with the powder size distribution in the experimental results. Furthermore, as the discharge power increases, the particle size distribution range becomes wider, and the average particle size becomes larger. Compared with changing the voltage and discharge time, changing the voltage has the most significant effect on the particle size range and average particle size. For the formation mechanism, the vapor film to droplet fragmentation theory was proposed to explain the reason of normal size distribution. During the process of droplet ejection, the vapor film formed by the boiling of the surrounding medium collapses at the lower temperature of the droplet. Then the large droplet breaks into small droplet with different sizes. 

## Figures and Tables

**Figure 1 materials-14-00287-f001:**
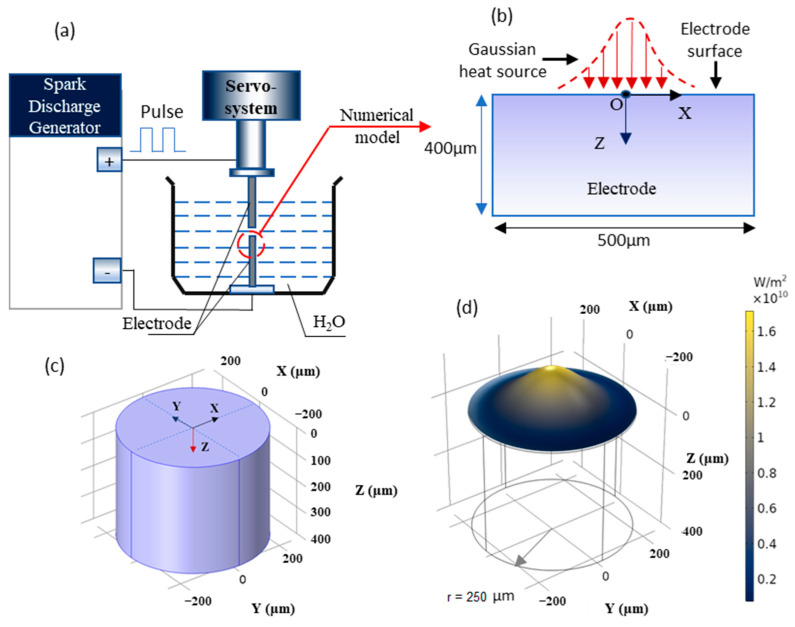
(**a**) The sketch of experimental device; (**b**) domain diagram of finite element model; (**c**) geometry for the numerical calculation; (**d**) gaussian thermal distribution (carried out by a current of 60 A and a voltage of 45 V).

**Figure 2 materials-14-00287-f002:**
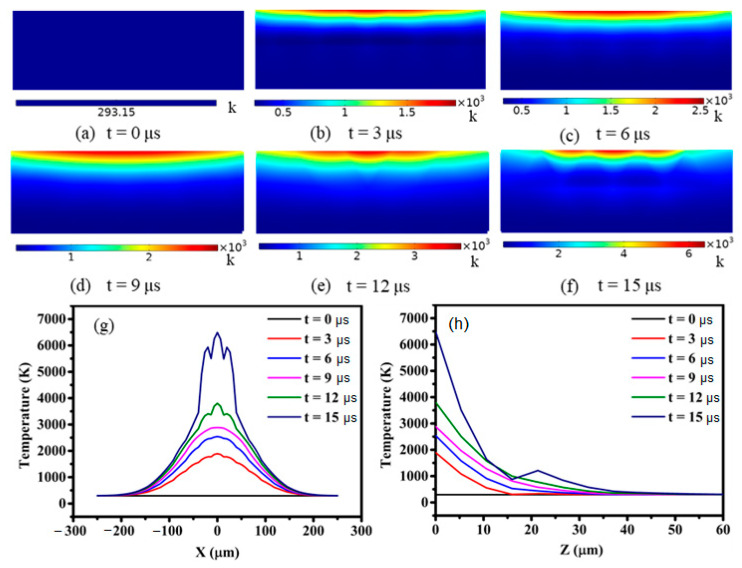
Temperature variation along the electrode surface (**a**–**f**) and depth direction with time-varying (**g**–**h**, calculated by current of 60 A and a voltage of 45 V).

**Figure 3 materials-14-00287-f003:**
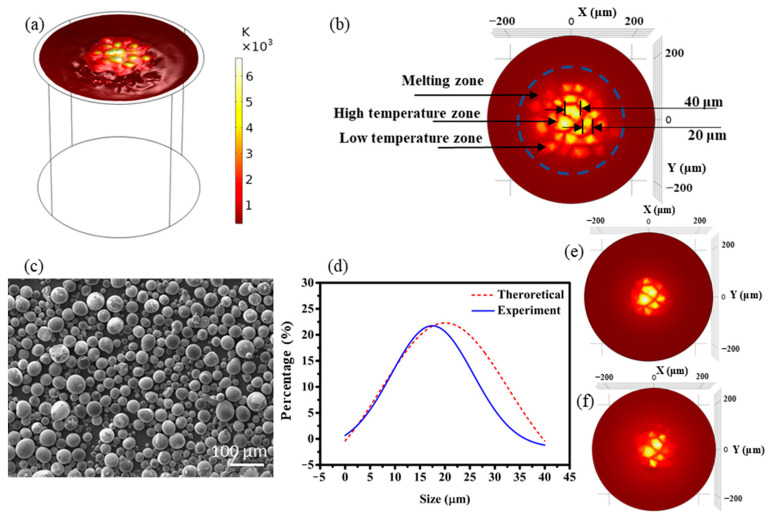
Relationship between heat distribution on electrode surface and particle size distribution. (**a**–**c**) 45 V, 60 A, 15 μs; (**d**) 45 V, 30 A, 15 μs; (**e**) 45 V, 45 A, 15 μs; (**f**) 45 V, 30 A, 15 μs.

**Figure 4 materials-14-00287-f004:**
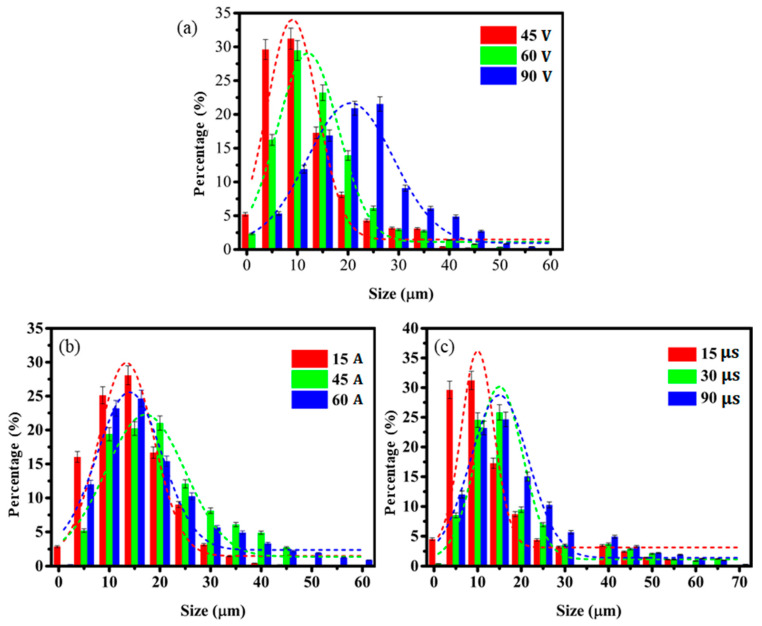
Effect of discharge parameters on particle size and SEM image. (**a**) voltage-particle size distribution; (**b**) current-particle size distribution; (**c**) discharge time-particle size distribution.

**Figure 5 materials-14-00287-f005:**
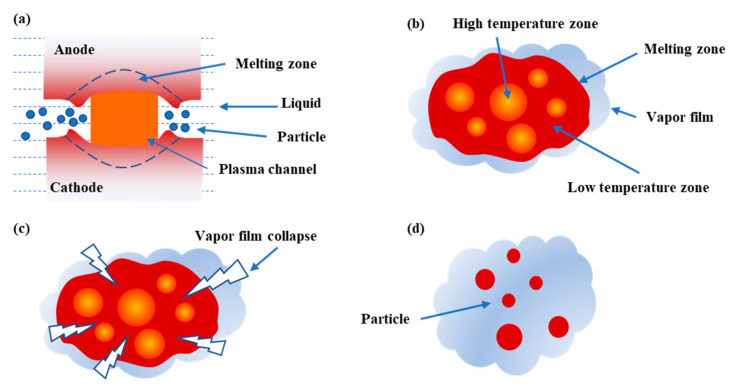
Schematic diagram of discharge and electrode thermal fragmentation process. (**a**) Schematic diagram of the spark discharge process; (**b**) film boiling around the droplet; (**c**) vapor film collapse; (**d**) particle formation.

**Table 1 materials-14-00287-t001:** Thermophysical properties of Ni [33].

Attribute	Value
Density	8908 kg/m³
Melting point	1455 K
Boiling point	2730 K
Latent heat of melting	298 kJ/kg
Latent heat of vaporization	6430 kJ/kg

**Table 2 materials-14-00287-t002:** Experimental conditions to generate nickel particles by the electrical discharge process.

Parameters	Group1			Group2			Group3		
	1	2	3	1	2	3	1	2	3
Voltage (V)	45	60	90	45	45	45	45	45	45
Current (A)	15	15	15	15	30	60	15	15	15
Pulse on (μs)	15	15	15	15	15	15	15	30	90
Pulse off (μs)	12.8								
Dielectric medium	Pure water								
Electrode	Nickel rods (purity better than 99.9%), density: 8.9 g/cm^3^								

## Data Availability

Data sharing is not applicable to this article.

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
