# Peer review of "The Numerical and Experimental Investigation of Particle Size Distribution Produced by an Electrical Discharge Process"

_materials, 2021, doi:10.3390/ma14020287_

Round 1
Reviewer 1 Report
In the present work, the authors developed the finite element multiphysics (FEM) coupling method to investigate the relationship between size distribution and thermal action.
Recommendation: Major revision
1. The manuscript is related to experiments and simulations work. Therefore, the title has to change. It might be "The numerical and experimental investigation of size distribution....."
2. Introduction is very insufficient. The manuscript is not up to the state-of-art.
3. Write the reference or references from where the thermo-physical properties of nickel were obtained.
4. Describe the finite element model schematically, e.g., shows direction and magnitude
5. The observations are presented in the result section. However, there is no detailed discussion on many obtained results.
Reviewer 2 Report
Title – what is “normal” means?
particles size distribution
The title should be re-considered
Please revise space between word and references (ref [1], ref [2-4], [8] - ALL of the references)
Please change “Sano [5] used” to “N. Sano [5] used” – please check all names authors (Jaworski, and other), please “et. all.” For co-authors
Bad English “However, a continuous Nitrogen should be supplied into the reactor” please change to (as possible version “However, a continuous supply of the N2 into the reactor should be ensured …” Since you are using chemical formulas please use a chemical symbol and formulas for all compounds (like ethanol, oxygen, nitrogen, water).
What is “pure water ” and is difference between with deionized water (please use H2O)
This only one example – English should be significantly improved.
Bad English “The size distribution of nickel powder” should be “The nickel powder particle size distribution”
What authors mean by “Then, a Gaussian thermal (Equation 1) was applied on the surface of the electrode.” ????
Bad eng “Table 1. Thermophysical properties of the three phases of nickel” maybe thermophysical properties of Ni ???
Table 2 according to the MDPI style no vertical lines in tables allowed.
Reviewer 3 Report
This article concerns nickel micro-powder fabricated by electrical discharge process and reports results of multivariant experiment in reference to numerical research. Authors presented and discussed well-documented relation between thermal action and size distribution of powder particles. Therefore they grouped technological parameters in three sets and precisely studied the influence of voltage, current, and discharge time on particle size distribution. Finally Authors showed formation mechanism of particle during electrical discharge. Summarizing, this paper has novelty and science soundness meet the requirements of Materials journal. Moreover the presented information can be very useful for people who are involved in powder metallurgy and additive manufacturing technologies. I strongly recommend this paper for publishing in Materials but after some minor correction, i.e.:
- Page 5, the last line – “…and time (45V and 15 us)…” please change “us” to “µs” – check throughout the article.
- Figure 2g,h – readers should be clearly informed what is X and Y, please add suitable titles for mentioned axes.
- Figure 3b,e,f – please add titles and units for all unnamed axes. The caption of Fig.3f is missing.
- Figure 4 – information about Figure 4a,b,c are missing in figure caption.
- Figure 5 – information about Figure 5a,b,c,d are missing in figure caption.
Round 2
Reviewer 1 Report
Accepted
Author Response
Thanks.
Reviewer 2 Report
The ethanol formula is C2H5OH, but (CH2OH)2 is ethylene glycol - please check appropriate use of it. Because in first manuscript version ethylene glycol wasn't mentioned please check.
I assume, that You are using 2 different ethanol formulas through the paper: CH3CH2OH and C2H6O(?) which actually al of the are correct, but it is very confusing, please use everywhere C2H5OH.
Actually to C2H6O correspond 2 substances -
- Dimethyl ether
- Ethanol
please clarify what is (C6H9NO)2 substance
As you are introduced a lot of chemical substances, I suggest the first time to give the trivial name (benzene, ethylene glycol, e.tc.) or IUPAC nomenclature name and provide a chemical formula (like you did in line 53 for WC), but by except water.
RE fig1.: very good and sufficient illustration, if this was taken from the literature, please provide the reference and please get the permission for use.
